# Real-World Insights into Efficacy and Safety of Enfortumab Vedotin in Japanese Patients with Metastatic Urothelial Carcinoma: Findings, Considerations, and Future Directions

Yuki Endo *, Jun Akatsuka ⬤, Hayato Takeda, Hiroya Hasegawa, Masato Yanagi, Yuka Toyama, Hikaru Mikami, Mikio Shibasaki, Go Kimura ⬤ and Yukihiro Kondo

Nippon Medical School, 1-1-5 Sendagi, Bunkyo-ku, Tokyo 113-8603, Japan; s00-001@nms.ac.jp (J.A.); s8053@nms.ac.jp (H.T.); h-hasegawa@nms.ac.jp (H.H.); area-i@nms.ac.jp (M.Y.); s4036@nms.ac.jp (Y.T.); h-mikami86@nms.ac.jp (H.M.); m-shibasaki@nms.ac.jp (M.S.); gokimura@nms.ac.jp (G.K.); kondoy@nms.ac.jp (Y.K.)
* Correspondence: y-endo1@nms.ac.jp; Tel.: +81-3-3822-2131

**Abstract:** This study presents the enfortumab vedotin (EV) treatment analysis at our institution. We retrospectively analyzed patients with metastatic urothelial cancer (mUC) treated with EV between January 2021 and October 2023. EV was administered at 1.25 mg/kg on days 1, 8, and 15 in a 28-day cycle. Whole-body computed tomography scans were performed to assess the treatment response. Patient characteristics, treatment histories, response rates, progression-free survival, and adverse events were evaluated. Response rates were determined, and adverse events were recorded. Among the 20 patients, 70% were male and 65% had bladder tumors. Most patients had lung (65%) or lymph node (65%) metastases. The median follow-up was 11.2 months, with 45% of the patients succumbing to the disease. The overall response rate was 55%. The median progression-free and median overall survivals were 10.5 and 12.9 months, respectively. Severe adverse events occurred in 35% of patients. In this real-world study, EV demonstrated promising efficacy and manageable safety profiles in Japanese patients with mUC. The study's results were consistent with previous clinical trials, although a longer follow-up was required. Our findings support EV use as a treatment option for patients with mUC who exhibit disease progression after platinum-based chemotherapy and immune-checkpoint inhibitor therapy.

**Keywords:** metastatic urothelial carcinoma; enfortumab vedotin; immune checkpoint inhibitors

## 1. Introduction

Bladder cancer is the 10th most commonly diagnosed cancer worldwide, with over 500,000 new cases reported annually [1]. Metastatic urothelial cancer (mUC) has a poor prognosis, and platinum-based chemotherapy (PBC) is the established standard of care. Unfortunately, the second-line chemotherapy has yielded disappointing results. In recent years, immune checkpoint inhibitors (ICIs) have emerged as promising treatment modalities, with five different ICIs approved for second-line use since 2016. Pembrolizumab has demonstrated superior outcomes compared with chemotherapy, and avelumab has been established as a maintenance treatment option [2,3]. Precision medicine has identified various therapeutic targets for bladder cancer, leading to the approval of erdafitinib for patients with fibroblast growth factor receptor (FGFR) alterations. However, eligibility for erdafitinib is limited to a small percentage of patients [4].

Despite the availability of these treatment options, mUC remains incurable [5]. To address this clinical challenge, enfortumab vedotin (EV), an antibody–drug conjugate comprising a monoclonal antibody targeting nectin-4 conjugated to the microtubule-disrupting agent monomethyl auristatin E was approved by the Food and Drug Administration (FDA)

in December 2019 for patients with mUC refractory to PBC and ICIs [6]. Initial FDA approval of EV was based on results from the EV-201 trial, while subsequent confirmation of their benefits for treatment-refractory mUC was obtained from a randomized phase 3 EV-301 trial, leading to full approval in July 2021 [7,8]. However, evidence supporting the efficacy and safety of EV is primarily derived from clinical trial data, with limited information available from real-world clinical practice.

To bridge this knowledge gap, we conducted a novel retrospective study to examine the therapeutic outcomes and safety profiles of EV in Japanese patients with mUC who were previously treated with PBC and ICIs. Importantly, our study stands as the first paper to provide single institution Japanese real-world data for this specific patient population. This unique perspective offers invaluable insights into the real-world clinical use of this treatment option within the Japanese context.

## 2. Materials and Methods

We retrospectively analyzed patients with mUC who were treated with EV between January 2021 and October 2023 and evaluated the efficacy and safety of EV in patients with mUC who received at least one dose of EV. The patient eligibility criteria included pathologically confirmed carcinoma of urothelial origin and the presence of metastatic disease.

EV was administered at a dose of 1.25 mg/kg on days 1, 8, and 15 at 28-day intervals. Dose reduction was allowed if the EV was intolerable, as determined by the treating physician. The Eastern Cooperative Oncology Group (ECOG) performance status, stage, histology, and EV-related toxicities were obtained from medical records.

Diagnostic monitoring of the tumor was performed at the start of EV treatment and every 1–3 months thereafter using chest, abdominal, and pelvic computed tomography. Treatment response was assessed using the Response Evaluation Criteria in Solid Tumors version 1.1, which included complete response (CR), partial response (PR), stable disease (SD), and progressive disease (PD). The safety profile of the patients was evaluated monthly using the National Cancer Institute Common Toxicity Criteria for Adverse Events, version 4.0. Our assessment of progression-free survival (PFS) and overall survival (OS) involved a comprehensive evaluation of multiple factors. These included analyzing complete blood count (CBC)/comprehensive metabolic panel (CMP) values, examining the status of cancer metastatic sites, assessing pre-existing diabetes before treatment, prior ICI (pembrolizumab or avelumab), and documenting adverse events (AEs). Furthermore, we conducted a comparative analysis between patients who received taxane therapy and those who did not.

PFS and OS were analyzed using statistical analysis. PFS was defined as the time from the start of EV treatment to PD. Further, OS was measured from the first day of EV exposure to the date of the last follow-up or death. Kaplan–Meier (K-M) curves were used to estimate survival distributions. We also gathered data on time elapsed from EV initiation to the observation of neuropathy exceeding Common Terminology Criteria for Adverse Events (CTCAEs) grade 2 (G2 neuropathy free-survival) and analyzed it using K-M curves. Statistical analyses and graph data were performed using SPSS software, version 25 (IBM Institute Corp., Armonk, NY, USA).

The study was approved by the Institutional Review Board of our institution (approval number [F3007-2]) and conducted in accordance with the Declaration of Helsinki. Written informed consent was obtained from all patients for mUC treatment with EV.

## 3. Results

Twenty patients with mUC previously treated with PBC and ICIs received EV monotherapy from January 2021 to October 2023 and were included in the analysis. The patient demographics and tumor characteristics are displayed in Table 1. Most patients were male (70.0%), with a median age of 73 years (range, 61–85 years). At the initiation of EV therapy, four patients (20%) had diabetes, two had Grade 2 neuropathy owing to prior therapy, seven (35.0%) had ECOG PS 1, seven (35.0%) had upper tract urothelial carcinoma (UTUC) as the primary tumor location, and eight (40.0%) had a pathological subtype. Lymph node

metastasis was present in 65% of the patients, while lung, liver, and bone metastases were observed in 65%, 15%, and 10% of the patients, respectively. Four prior regimens were documented in 25% of the patients, three regiments in 30%, and two regimens (PBC and ICI) in 45%. The EV dose was adjusted based on the patient's general condition and tolerability, resulting in a dose reduction in 50% of patients during the treatment period.

**Table 1.** Background and characteristics of the patients (*n* = 20).

| Background of the Patients | | | |
|---|---|---|---|
| sex | | *n* | % |
| | female | 6 | 30.0% |
| | male | 14 | 70.0% |
| age | | median | range |
| | | 73 | 61–85 |
| follow up months | | | |
| | | 8.2 | 1.4–20.8 |
| ECOG-performance status | | *n* | % |
| | 0 | 13 | 65.0% |
| | 1 | 7 | 35.0% |
| primary location | | | |
| | UTUC | 7 | 35.0% |
| | LTUC | 13 | 65.0% |
| pathological subtype (variant) | | | |
| | yes | 8 | 40.0% |
| | no | 12 | 60.0% |
| metastatic locations | | | |
| | lymph node | 13 | 65.0% |
| | lung | 13 | 65.0% |
| | liver | 3 | 15.0% |
| | peritonium | 0 | 0.0% |
| | bone | 2 | 10.0% |
| | local | 0 | 0.0% |
| number of prior regimens | | | |
| | 2 | 9 | 45.0% |
| | 3 | 6 | 30.0% |
| | 4 | 5 | 25.0% |
| prior immune checkpoint inhibitor | | | |
| | pembrolizumab | 10 | 50% |
| | avelumab | 10 | 50% |
| prior taxane therapy | | | |
| | yes | 4 | 20% |
| | no | 16 | 80% |

ECOG: Eastern Cooperative Oncology Group.

The outcomes of this study are summarized in Table 2. The median follow-up period was 11.2 months (range, 2.7–22.5 months), and the median number of administration cycles was 7 (range, 2–18 cycles) (Figure 1). All the patients had measurable tumor masses. The maximum shrinkage of the target lesions during the treatment period in our cohort is depicted in Figure 2. As the best response, two patients (10%) had CR, nine (45%) had PR, eight (40%) had SD, and one (5%) had PD. Thus, 11 patients (55%) had an objective response (CR + PR), and 15 patients (95%) had disease control (CR + PR + SD). During the treatment period, five patients (25%) died (all deaths were cancer-related). The median PFS was 10.5 months, and the median OS was 12.9 months (Figure 3a,b). The analysis of CBC/CMP did not reveal any statistically significant differences in hemoglobin levels, neutrophil-to-lymphocyte ratio, estimated glomerular filtration rate, or albumin levels (Supplementary Figures S1–S4).

**Table 2.** Outcomes of the patients (*n* = 20).

| Outcome of the Patients | | |
|---|---|---|
| | median | range |
| follow up months | 11.2 | 2.7–22.5 |
| administration cycles | 7 | 2–18 |
| | *n* | % |
| best response | | |
| complete response | 2 | 10.0% |
| partial response | 9 | 45.0% |
| stable disease | 8 | 40.0% |
| progression disease | 1 | 5.0% |

## Efficacy of Enfortumab Vedotin (swimmers plot)

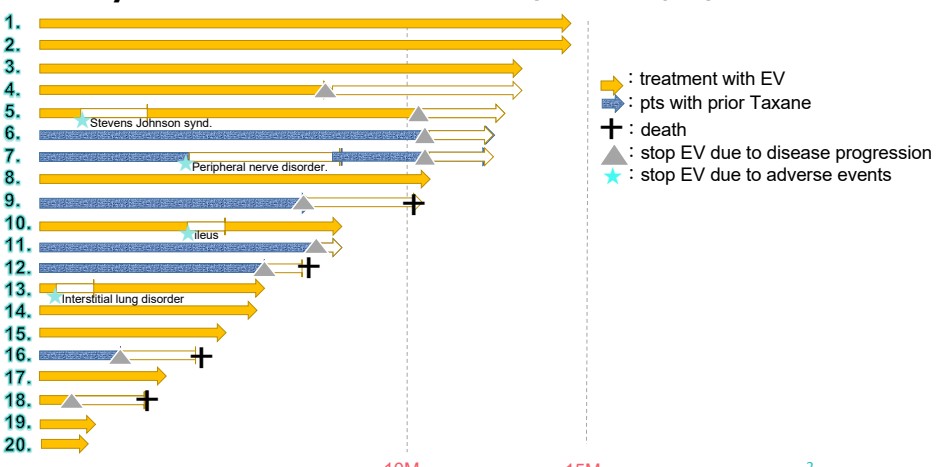

**Figure 1.** Swimmer plot of 20 patients. The yellow or blue arrow shows patient survival with the continuation of enfortumab vedotin (EV) treatment, and the white arrow indicates patient survival with discontinuation. The gray triangle displays the termination of EV treatment due to disease progression. A blue star indicates termination of EV treatment due to adverse events. A cross mark indicates patient death.

## Efficacy of Enfortumab Vedotin (water fall plot)

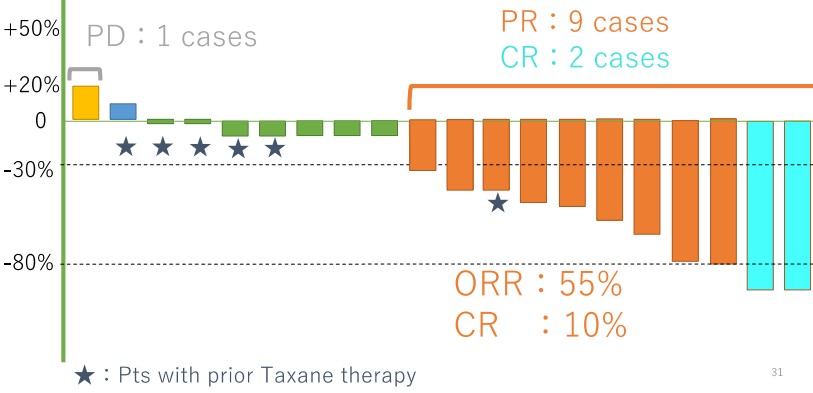

**Figure 2.** Waterfall plot of 20 patients. Each line shows the tumor size when the best response was obtained compared with the baseline calculated based on RECIST v1.1. Star marks patients previously treated with Taxane therapy. Sax blue lines showed complete response, Orange showed partial response, green and dark blue showed stable disease, and yellow showed progression disease.

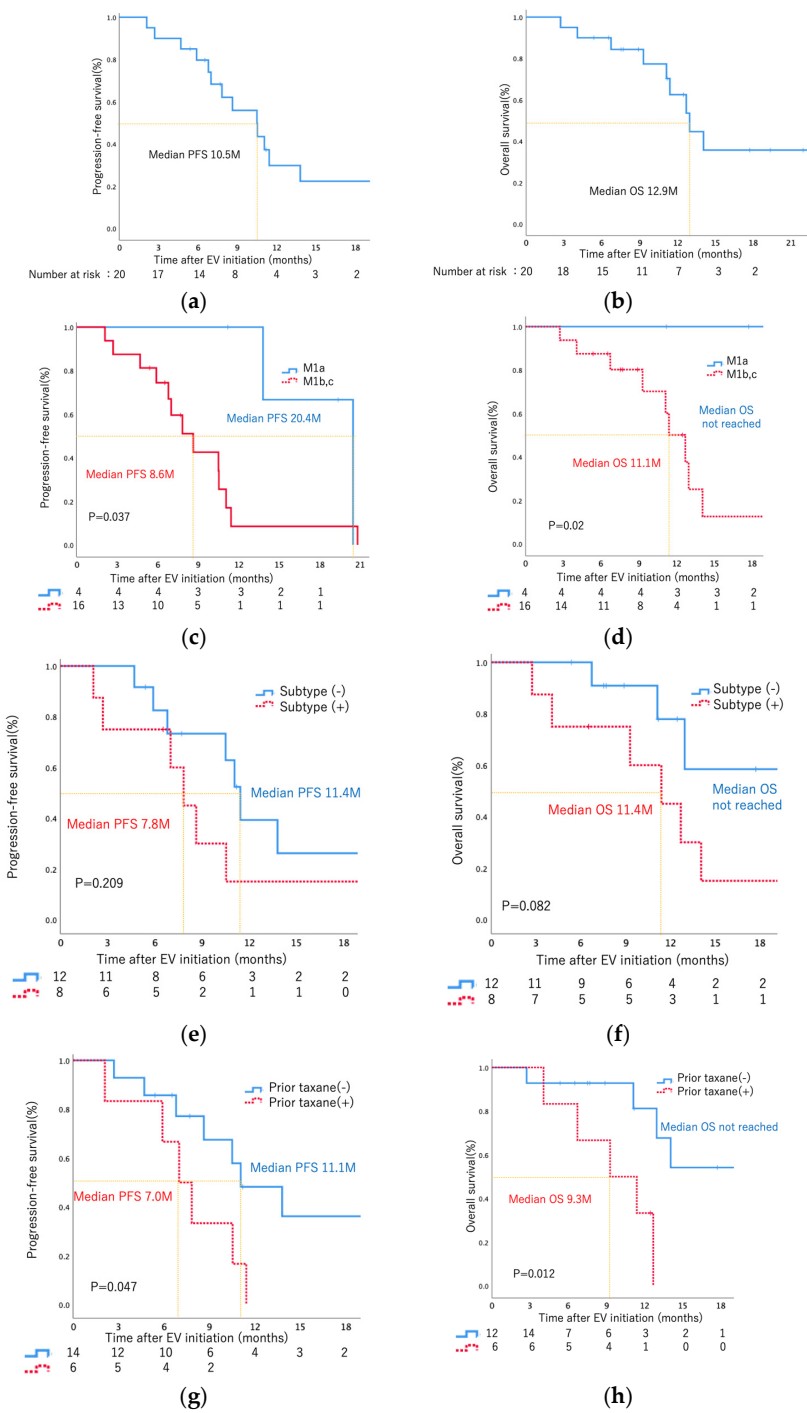

**Figure 3.** (**a**) Kaplan−Meier curve of patient progression−free survival rate. (**b**) Kaplan−Meier curve of overall survival rate. (**c**) Kaplan–Meier curve comparing the progression−free survival rate of patients who had metastasis limited to lymph nodes (M1a) versus metastasis involving other organs (M1b, M1c). (**d**) Kaplan−Meier curve comparing the overall survival rate of patients who had metastasis limited to lymph nodes (M1a) versus metastasis involving other organs (M1b, M1c). (**e**) Kaplan–Meier curve comparing the progression−free survival rate of patients who had subtype histology with those who did not. (**f**) Kaplan−Meier curve comparing the overall survival rate of patients who had subtype histology with those who did not. (**g**) Kaplan–Meier curve comparing the progression-free survival rate of patients who had previously received taxane with those who had not. (**h**) Kaplan−Meier curve comparing the overall survival rate of patients who had previously received taxane with those who had not.

When examining metastatic sites, a notable distinction emerged in PFS and OS between patients with limited lymph node metastasis (M1a) and those with metastasis in other organs (M1b, M1c) (PFS: 20.4 months vs. 8.6 months, log-rank $p = 0.037$; OS: not reached vs. 11.1 months, log-rank $p = 0.02$) (Figure 3c,d). However, no statistically significant differences in either PFS or OS were observed among patients with liver and bone metastasis (Supplementary Figures S5 and S6) or prior ICI treatment (Supplementary Figure S7). Similarly, no significant difference in PFS or OS was observed between patients, positive or negative, for the specific histological subtype (Figure 3e,f). The specific details regarding subtype histology for eight patients can be found in Supplementary Table S1.

Notably, patients without prior taxane therapy exhibited significantly longer PFS and OS compared to those with prior taxane therapy (PFS: 11.1 months vs. 7.0 months, log-rank $p = 0.047$; OS: not reached vs. 9.3 months, log-rank $p = 0.012$) (Figure 3g,h).

All patients were evaluated for toxicity, and the AEs experienced during treatment are summarized in Table 3. All patients in our study experienced treatment-related AEs, most of which were of mild-to-moderate severity (grade 1/2). The most common AEs caused by EV administration were dermatosis ($n = 17$; 85%), peripheral sensory neuropathy ($n = 16$; 80%), and dysgeusia ($n = 13$; 65%). Overall, 11 severe AEs (grade 3 or 4) occurred in seven patients (35%) (peripheral sensory neuropathy in five [25%], loss of appetite in two [15%], fatigue in one [5%], neutropenia in one [5%], anemia in one [5%], and interstitial lung disease in [ILD] one [5%]). Due to EV toxicity, seven patients (35%) required a dose reduction or interruption; however, none required drug discontinuation due to AEs.

**Table 3.** Safety of enfortumab vedotin (adverse events).

| Adverse Event | | | | |
|---|---|---|---|---|
| | G1-2 | G3-4 | total | % |
| dermatosis | 17 | 0 | 17 | 85% |
| peripheral sensory neuropathy | 11 | 5 | 16 | 80% |
| dysgeusia | 13 | 0 | 13 | 65% |
| loss of appetite | 9 | 2 | 11 | 55% |
| fatigue | 10 | 1 | 11 | 55% |
| alopecia | 10 | 0 | 10 | 50% |
| anemia | 4 | 1 | 5 | 25% |
| liver enzyme elevation | 4 | 0 | 4 | 20% |
| blurred vision | 3 | 0 | 3 | 15% |
| diarrhea | 3 | 0 | 3 | 15% |
| interstitial lung disease | 2 | 1 | 3 | 15% |
| neutropenia | 0 | 1 | 1 | 5% |

For neuropathy-free survival (NFS), two patients experienced neuropathy due to prior therapy. To adjust for their pre-existing neuropathy at baseline, the K-M curve commenced from 90% on day 0. The median NFS was 4.1 months (Supplementary Figure S8). Subsequently, we assessed grade 2 (G2) NFS after the administration of EV based on the patient's history of diabetes mellitus (DM). Patients with DM exhibited a longer G2 NFS compared to those without DM (5.7 months vs. 3.3 months, $p = 0.048$) (Supplementary Figure S9).

## 4. Discussion

EV is approved for treating advanced UC in patients who are refractory to both PBC and ICIs. This recommendation is supported by the success of the EV-301 trial [8]. Despite its established clinical efficacy and approvals across major regions, real-world data on EV are limited. This retrospective study is the first from a single institution to explore the real-world efficacy and safety of EV. The study delved into patient profiles, including cancer subtypes, prior treatments, response rates, survival outcomes, and safety.

The overall response rate (ORR) for EV therapy was 55% in our group, which is better than the 40.6% in the EV-301 trial. The median OS was 12.9 months for our cohort, closely aligning with the 12.88 months reported in the trial [8]. A previous retrospective registration

study (UNITE) recruited 260 patients from 16 academic institutions in the United States and demonstrated that the ORR was 52%, and the median PFS and OS from the start of EV were 6.8 and 14.4 months, respectively [9]. In two other multicenter retrospective studies from Japan, Taguchi et al. reported that the ORR was 46%, and the median PFS and OS from the start of EV were 5 and 11 months, respectively, in 39 patients from nine institutions. Additionally, Miyake et al. reported that the ORR was 56% (CR, 0%; PR, 0%), and the median PFS and OS from the start of EV were 9 and 16 months, respectively, in 34 patients from 19 institutions [10,11]. These retrospective studies concluded that the efficacy of EV therapy in real-world settings was comparable to that reported in clinical trials. These consistent results underscore the efficacy of EV in Japanese patients with mUC.

However, there are notable disparities in patient demographics among the EV301 trial, UNITE study, and our study. In the EV-301 trial, patients with UC-containing squamous differentiation or multiple cell types were included, while those with UC-containing variant histology were excluded [8]. The UNITE study enrolled both platinum-pretreated and platinum-naïve patients, with 68% having pure UC; however, mixed subtypes predominated (2%), and pure subtypes were also part of the study (1%). Notably, the response rate for pure UC reached 58%, contrasting with 42% in patients with subtype histology [9]. In these two Japanese reports, subtype information is not available [10,11]. In our study, the response rate among patients with histological subtypes was 25% (2/8), with a response rate of 75% in those with pure UC, mirroring the UNITE study [9]. These findings suggest a robust response rate of EVs to histological subtypes; however, it is lower than that of pure UC. This aligns with prior reports indicating lower Nectin-4 expression in tissues containing rare subtype histology than in pure UC [12]. Additionally, we observed one case of disease progression in a patient with an 80% plasmacytoid subtype, hinting at a potentially poorer prognosis associated with this histological subtype. Subsequently, we conducted an examination of PFS and OS according to subtype histology, revealing no statistically significant differences, which might be attributed to the small cohort. The consistency of subtype data from a single institution and by a single pathologist is highly valuable.

Moreover, 30% of our cohort had previously undergone taxane chemotherapy. EVs, nectin-4-directed antibodies conjugated to monomethyl auristatin E (MMAE), share microtubule dynamics disruption similarities with taxanes [7,8,13]. Thus, EV therapy might be less effective in patients previously treated with taxanes. Our study demonstrated significantly longer PFS and OS in patients without prior taxane therapy compared to those with prior taxane therapy (PFS: 11.1 months vs. 7.0 months, log-rank $p = 0.047$; OS: not reached vs. 9.3 months, log-rank $p = 0.012$). However, previous reports by Miyake et al. highlighted two cases where EV was effective in patients previously treated with taxanes [13]. Notably, EV therapy in these cases commenced more than 6 months after the final taxane treatment. Conversely, all taxane treatments in our study occurred within 6 months before EV therapy initiation. These findings suggest that administering EV therapy to patients previously treated with taxanes should be delayed until at least 6 months after the last treatment.

We encountered two cases of CR, both of which had no prior taxane therapy and metastasis limited to lymph nodes (M1a). Additionally, in our cohort, PFS and OS for patients with metastasis limited to lymph nodes (M1a) versus metastasis involving other organs (M1b, M1c) were significantly longer than those with organ-confined metastasis (PFS: 20.4 months vs. 8.6 months, log-rank $p = 0.037$; OS: not reached vs. 11.1 months, log-rank $p = 0.02$). These findings suggest that EV therapy may be more effective for patients with metastasis limited to lymph nodes compared to those with metastasis involving other organs.

In our study, similar to the EV-201 and EV-301 trials, all patients experienced treatment-related AEs. In the EV-201 trial, 19% experienced serious (grade 3–4) AEs, and 12% had AEs leading to treatment discontinuation, with no treatment-related deaths reported [7]. The EV-301 trial also reported that 19% of participants experienced serious AEs, 13.5% had AEs that led to treatment discontinuation, and 2.4% experienced AEs that led to death [8]. Miyake

et al.'s multi-institutional study reported that 76% of patients experienced any-grade AEs, with 24% of serious AEs and 15% of AEs leading to the discontinuation of EV therapy [11]. In our study, with 35% of serious AEs and no AEs leading to treatment discontinuation, we observed a higher overall incidence of AEs compared to Miyake et al.'s report, primarily due to AEs resulting from prior subsequent therapies. Specifically, 45% of our patients had received two prior regimens, 30% received three, and 25% received four prior treatments. Additionally, in our study, the median NFS was 3.9 months, suggesting that patients were likely to develop neuropathy approximately 4.1 months after EV administration. These findings underscore the significance of effectively managing AEs during EV therapy for Japanese patients with mUC and emphasize the need for further research and real-world data collection to gain a deeper understanding of the safety aspects associated with this treatment option.

However, we observed one case of grade 3 ILD in our study, which occurred 1 month after the onset of EV. The possibility of an immune-related AE (irAE) due to prior treatment with avelumab cannot be ruled out. The EV301 trial did not report ILD as a treatment-related AE because patients with ongoing clinically significant toxic effects and irAEs related to prior therapy were excluded. However, in real-world treatments, the sequence of PBC and ICI is mandatory, and ILD may occur in that sequence, suggesting its frequent occurrence. Yoon et al. retrospectively examined 64 Koreans who participated in the EV-201 and EV-301 trials, of whom 18 (28.1%) developed all grades of EV-associated pneumonia, and 2 (11.1%) died [14]. In our study, one patient who developed ILD resumed EV therapy after drug withdrawal. However, clinicians should closely monitor patients who experience immunotherapy failure for ILD development.

## 5. Conclusions

The study has certain limitations, including a small sample size, a short follow-up period, and a lack of a control group. Additionally, our study included patients from only one institution and may not be representative of the broader population. Furthermore, the study's retrospective nature and reliance on medical records may have resulted in incomplete or inaccurate data. Finally, the possibility of selection bias cannot be ruled out, as the patients were included based on specific criteria. Therefore, these limitations should be considered when interpreting the results of this study.

This real-world study provided valuable insights into the efficacy and safety of EV in Japanese patients with mUC. The ORR and disease-control rates were consistent with those of previous studies, while the PFS and OS were longer owing to the short follow-up period. However, close monitoring for interstitial lung disease is essential and caution should be exercised when administering EV to patients previously treated with taxanes. These findings contribute to understanding the real-world effectiveness of EVs and aid in identifying optimal patient populations for future clinical use. However, further research with larger sample sizes and longer follow-up periods is warranted to validate these findings.

**Supplementary Materials:** The following supporting information can be downloaded at https://www.mdpi.com/article/10.3390/curroncol31020056/s1, Figure S1: Kaplan–Meier curve comparing progression-free survival (PFS) and overall survival (OS) in patients categorized by the presence or absence of anemia; Figure S2: Kaplan–Meier curve comparing progression-free survival (PFS) and overall survival (OS) in patients categorized by a Neutrophil-to-Lymphocyte Ratio (NLR) of more than 2.5 or less; Figure S3: Kaplan–Meier curve comparing progression-free survival (PFS) and overall survival (OS) among patients categorized by a glomerular filtration rate (GFR) exceeding the median value and those below it; Figure S4: Kaplan–Meier curve comparing progression-free survival (PFS) and overall survival (OS) in patients categorized by a albumin of more than 4.0 mg/dl or less; Figure S5: Kaplan–Meier curve comparing progression-free survival (PFS) and overall survival (OS) in patients categorized by the presence or absence of liver metastasis; Figure S6: Kaplan–Meier curve comparing progression-free survival (PFS) and overall survival (OS) in patients categorized by the presence or absence of bone metastasis; Figure S7: Kaplan–Meier curve comparing progression-free survival (PFS) and overall survival (OS) in patients categorized by the prior immune-checkpoint

inhibitors; Figure S8: Kaplan–Meier curve of patient G2 neuropathy-free survival rate; Figure S9: Kaplan–Meier curve comparing the G2 neuropathy-free survival rate of patients who had past history of diabetes meritus with those who did not; Table S1: Details of subtypes (*n* = 8).

**Author Contributions:** Conceptualization, Y.E. and Y.K.; methodology, Y.E. and M.S.; software, Y.E. and Y.E.; validation, J.A., G.K. and Y.T.; formal analysis, Y.E., H.T. and H.H.; investigation, Y.E., J.A., H.T., H.H., M.Y., Y.T., H.M. and M.S.; resources, G.K. and Y.K.; data curation, Y.E., J.A., H.T., H.H., M.Y., Y.T., H.M. and M.S.; writing—original draft preparation, Y.E.; writing—review and editing, Y.K., Y.T., H.M. and M.S; visualization, Y.E., M.Y., H.M. and M.S.; supervision, J.A., G.K. and Y.K.; project administration, Y.E., H.T, G.K. and Y.K. All authors have read and agreed to the published version of the manuscript.

**Funding:** This research received no external funding.

**Institutional Review Board Statement:** The study was conducted in accordance with the Declaration of Helsinki and approved by the Nippon Medical School ethical committee (approval number [FR0208-39]).

**Informed Consent Statement:** Informed consent was obtained from all participants involved in the study.

**Data Availability Statement:** The data underlying this article will be shared on reasonable request to the corresponding author.

**Acknowledgments:** The authors are grateful to K. Obayashi for suggesting the topic treated in this paper and for useful discussion as well.

**Conflicts of Interest:** Yukihiro Kondo had personal payment or honoraria for lectures, presentations, speakers' bureaus, manuscript writing, or educational events from the following companies: Astellas Pharma Inc., Nippon Kayaku Co., Ltd., and Bristol Meyers Squibb. Go Kimura had personal payment or honoraria for lectures, presentations, speakers' bureaus, manuscript writing, or educational events from the following companies: Ono Pharmaceutical, Bristol Myers Squibb, Merck Biopharma, Janssen Pharmaceutical, MSD, Eisai, Takeda, and Astellas. Yuki Endo had personal payment or honoraria for lectures, presentations, speakers' bureaus, manuscript writing, or educational events from the following companies: Astellas Pharma Inc., Nippon Kayaku Co., Ltd., Ono Pharmaceutical, and MSD.

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
