# Peer review of "Real-World Insights into Efficacy and Safety of Enfortumab Vedotin in Japanese Patients with Metastatic Urothelial Carcinoma: Findings, Considerations, and Future Directions"

_curroncol, doi:10.3390/curroncol31020056_

Round 1

Reviewer 1 Report

Comments and Suggestions for Authors

This study investigates the application of novel pharmacological interventions in the treatment of metastatic urothelial carcinoma, a cancer characterized by low survival rates, thereby underscoring its significance. However, there are several critical issues within this research:

  1. 1. The small sample size undermines the reliability of the study's conclusions.
  2. 2. Patient inclusion was conducted up until April 2023, which raises concerns regarding the adequacy of the follow-up period for sufficiently assessing survival outcomes.

Author Response

Thank you for your insightful feedback. In response to the valuable comments and suggestions provided by the reviewers, we have diligently revised the manuscript to address the raised points and improve the overall quality of the study.

Reviewer 2 Report

Comments and Suggestions for Authors

Endo et al presents a small single-institution retrospective study of enfortumab vedotin monotherapy in the 3rd+ line setting for 20 patients with metastatic UC. They point out this is a unique study because it focuses on a Japanese population only. 

What has been presented is on the simpler side, which is likely fine for this journal, but the paper can be strengthened if the following is addressed.

It would be much more interesting to see more analyses done. A few recommendations include:

For background characteristics, it would be interesting to note who already had underlying conditions that EV is known to exacerbate (such as neuropathy and diabetes).

Since the study is small, please specifically comment more on the details of the 2 patients with CR and the 1 patient with PD. What are the hypotheses that led to the CR vs PD for these patients?

Any genetic tumor profiling of the main tumor/metastatic site and CBC/CMP values of interest (such as Hgb, neutrophils, lymphocytes, creatinine clearance, albumin, etc.) for any of these patients? Predictive factors to EV are lacking, and if there are any comparisons that can be made for those who progressed quickly vs those who had a good response would be worth investigating. This should also be addressed in the discussion.

For patients who had an adverse event to EV, how these patients ended up doing - did they have a better PFS/OS?

For the patients with different metastatic sites prior (as shown in Table 1), how did patients with liver mets and bone mets respond to EV, compare to patients with lymph node mets, etc.

Comments on the Quality of English Language

Enfortumab vedotin does not have a hyphen - please remove throughout the article. Syntax needs to be checked. In line 12, -301 is missing EV before it.

Author Response

(The authors gave the same response as above.)

Reviewer 3 Report

Comments and Suggestions for Authors

The manuscript is well written and well structured. I did not detect any significant critical issues; however, it would be useful to specify the subtype of histological variant, which the authors do not specify

Author Response

(The authors gave the same response as above.)

Round 2

Reviewer 2 Report

Comments and Suggestions for Authors

Authors claim they are the first Japanese study to evaluate EV in 3rd line setting in the real world. I do not believe this is the case. I have found two studies published.

PMID: 38061911, 37686503

Please cite both studies and compare and contrast your findings with these published studies' findings in your discussion. Please also remove the statement of being first in the last paragraph of the introduction.

Fig 1 needs to be more professionally made. In addition, interstitial and disorder are spelled incorrectly.

Comments on the Quality of English Language

Minor spelling errors are noted.

Author Response

Dear Reviewer 2

Thank you for your thoughtful and constructive feedback on our manuscript titled "Real-World Insights into Efficacy and Safety of Enfortumab Vedotin in Japanese Patients with Metastatic Urothelial Carcinoma: Findings, Considerations, and Future Directions" which we submitted to Current Oncology

In response to your comments, we have made the following changes to our manuscript, as detailed in the attached Word file. We kindly request you to refer to the document for a comprehensive view of the revisions.

We have presented only the revised sections of the paper this time. As suggested by the academic editor, the entire manuscript is currently undergoing further English language editing. Once this process is complete, I would like to upload the revised manuscript again.

We greatly appreciate your valuable input, which has undoubtedly improved the quality of our manuscript. If you have any further comments or suggestions, please do not hesitate to share them with us. We look forward to your continued guidance.

Once again, thank you for your time and expertise.
